# Female sex worker preferences for HIV pre-exposure prophylaxis delivery in Uganda: A discrete choice experiment

Ruth Mpirirwe[1,2]*, Rita Makabayi-Mugabe[3], Laban Muteebwa[1], Onesmus Kamacooko[1], Felix Wamono[2], Mayanja M. Kajumba[4], Joan Nangendo[1], Fred C. Semitala[1], Peter Kyambadde[5], Katumba James Davis[1], Joan Kalyango[1], Charles Karamagi[1], Agnes Kiragga[5], Mari Armstrong-Hough[6], Sarah E. G. Moor[7], Anne R. Katahoire[1,4], Moses R. Kamya[1], Andrew Mujugira[3,8]

**1** Department of Clinical Epidemiological Unit, Makerere University, Kampala, Uganda, **2** Department of Statistical Methods and Actuarial Sciences, Makerere University, Kampala, Uganda, **3** Infectious Diseases Institute, Kampala, Uganda, **4** Departmentof Clinical Psychology, Makerere University, Kampala, Uganda, **5** Department Sexually Transmitted Diseases/AIDS Control Programm, Ministry of Health, Kampala, Uganda, **6** Department of Global Public Health, New York Univeristy, NewYork, NewYork, United States of America, **7** Department of Pediatrics, University of Toronto, Toronto, Ontario, Canada, **8** Department of Global Health, University of Washington, Seattle, Washington, United States of America

* ruthmpirirwe@gmail.com

## Abstract

Cisgender female sex workers (FSWs) in sub-Saharan Africa have a high risk of HIV acquisition, highlighting the need for innovative approaches to expand coverage of evidence-based HIV prevention methods, including oral pre-exposure prophylaxis (PrEP). Our study aimed to identify FSWs' preferences for a PrEP delivery model with structured choices for delivery location, services offered, and adherence support. We conducted a discrete choice experiment (DCE) with female sex workers (FSWs) aged 18 and above at the Most At-Risk Population Initiative (MARPI) clinic in Kampala, Uganda, from October to November 2023. Participants were recruited consecutively. To identify the most preferred PrEP delivery model, we designed eight choice sets using a D-efficient design. Each set included three PrEP service options and an opt-out. Options varied by provider, delivery location, channel, and support services. Participants selected their preferred option in each set. Preferences and trade-offs were analyzed using a panel mixed model, and the highest median utility identified the top model. Overall, 203 participants completed the DCE. The median age was 24 years (interquartile range [IQR] 20–32). Most FSWs preferred receiving PrEP from a healthcare worker at the clinic with short message service (SMS) reminders for adherence support (median utility score 0.87; interquartile range [IQR] 0.82, 0.94). This preference remained consistent across all age groups, with a median utility score of 0.88 for ages 15–19, 0.87 for ages 20–24, and 0.85 for ages ≥25. FSWs preferred to receive PrEP care directly from providers at healthcare facilities and highlighted the need for additional

**Data availability statement:** All relevant data are within the paper and its Supporting Information files.

**Funding:** Fogarty International Center, National Institute of Alcohol Abuse and Alcoholism, National Institute of Mental Health, of the National Institutes of Health under Award Number D43 TW011304, and Makerere University Research and Innovations Fund (MakRIF) under award Number PhD round 4 supported the research reported in this publication. The funders had no role in study design, data collection and analysis, decision to publish, or preparation of the manuscript. The content is solely the authors' responsibility and does not necessarily represent the official views of the National Institutes of Health.

**Competing interests:** The authors have declared that no competing interests exist.

support in the form of SMS reminders to improve adherence and prevent HIV acquisition. This preferred model, if implemented, could increase prevention coverage and inform future approaches to delivering PrEP through the Uganda National PrEP Program.

## Introduction

Cisgender female sex workers (FSWs) are at high risk of HIV acquisition but have limited access to effective HIV prevention interventions [1]. This underscores the urgency for innovative strategies to increase the uptake of evidence-based biomedical HIV prevention methods, including oral pre-exposure prophylaxis (PrEP) [2,3]. However, a one-size-fits-all approach to HIV prevention may not effectively address the diverse needs and preferences of key populations. Discrete choice experiments (DCEs) have revealed diverse stated preferences for HIV prevention within individuals and communities [4]. Person-centered care models may overcome these barriers, but they are inconsistently implemented for key populations globally [5,6]. Therefore, it is imperative to better understand how to effectively implement person-centered PrEP care and optimize HIV prevention delivery approaches.

Uganda has established facility and community models for PrEP delivery, consisting of four main components: the target population, infrastructure for providing PrEP, trained PrEP providers, and designated delivery channels. However, these models do not consider FSWs' unique needs and preferences [7]. As a result, FSWs who receive PrEP through the facility model often face challenges such as long travel distances to the clinic and lengthy waiting times. This can lead to significant direct costs, such as transportation expenses, and indirect costs, like loss of work time. These barriers hinder their ability to adhere to and remain in PrEP care [8–10]. Despite being an effective biomedical intervention for reducing HIV transmission when taken correctly, PrEP persistence among FSWs in Uganda is low [11]. Given the high HIV prevalence (37%) [12] among FSWs in Uganda, it is crucial to understand which PrEP delivery model would best meet their specific needs and promote effective PrEP use and retention in care to decrease HIV incidence [13,14].

Prior research has emphasized incorporating choice within person-centered models for HIV service delivery [15]. This includes providing additional support for medication adherence and offering flexibility in clinic-based or off-site delivery. Despite this, there is currently limited literature documenting the specific preferences of FSWs for PrEP delivery options [8]. With the expansion of PrEP programs, it is crucial to establish effective and feasible delivery models to ensure maximum coverage [16,17]. To address these gaps, our study sought to identify Ugandan FSW preferences attending the MARPI clinic for a PrEP delivery model that offered structured options for location, services provided, and adherence support.

## Materials and methods

### Study population and setting

We conducted a discrete choice experiment (DCE) at the Most At-Risk Population Initiative (MARPI) clinic within the Mulago National Referral Hospital complex in Kampala, Uganda, from October to November 2023. A discrete choice design is a research method used to understand and predict how people make decisions when faced with a set of alternatives. We recruited 203 FSWs from the MARPI clinic, which serves an estimated 10,000 FSWs annually. Before participation in the DCE, FSWs received information about the study's objectives and procedures and were asked to provide informed consent. They were informed that the DCE was a quantitative research method that explored individual preferences by presenting hypothetical options and evaluating responses to specific program, product, or service attributes [18]. Eligibility criteria included being ≥18 years, on PrEP for at least two months, consent to participate in the study, selling sex within Kampala, Mukono, or Wakiso districts in Central Uganda, and receiving PrEP refills at the MARPI clinic. Participants were excluded from the study if they met the following criteria: currently participating in another PrEP or HIV prevention study, allergic to tenofovir, lamivudine, emtricitabine, or other PrEP medication, being infected with Hepatitis B virus or having chronic kidney disease (based on self-report or medical records), as these individuals would eventually be discontinued from PrEP.

### Attributes and levels

We conducted a systematic review and qualitative study to identify DCE attributes [19,20]. These findings were used to inform our study's final selection of attributes (Table 1). To create the choice sets, we employed a fractional factorial design; fractional D-deficiency designs are valuable for optimizing experimental efficiency when full factorial designs are not feasible. We relied on assumptions about factorial structure, D-efficiency, balance, confounding, additivity, variance homogeneity, and sample size, which resulted in eight sets (Table 2). Previous studies have established that more than eight choice tasks can impose cognitive and time limitations on participants [21,22]. We also included an opt-out response option where respondents could choose "neither" to indicate their dissatisfaction with the provided PrEP delivery models. This resulted in a final design with a D-efficiency of 88.4%.

Table 1. DCE attributes and levels based on a systematic review, qualitative findings, and expert panel review.

| No. | Attribute | Level | Attribute definition for this study |
|---|---|---|---|
| 1. | PrEP provider | 1. Health worker | This can be a doctor, nurse, PrEP counselor, or other healthcare provider |
| | | 2. Peer | An FSW who is on PrEP and serves as a role model to the other FSW |
| | | 3. None | No preference for either |
| 2. | Place of PrEP delivery | 1. Home | A place where the FSW lives |
| | | 2. Community | A designated place in the community where health workers come and offer HIV testing services and drug refills |
| | | 3. Health facility | Health care setting that provides PrEP |
| | | 4. Hotspot | A specific location where FSWs gather and transact sex |
| 3. | Delivery channel | 1. In-person visits | FSW picks their own PrEP drugs |
| | | 2. Family member | A brother, spouse, sister, or member of the extended family picks the drug |
| | | 3. Peer | An FSW who is on PrEP and serves as an example to the other FSW |
| 4. | Additional support | 1. Phone call reminder | A text message sent to a mobile phone 30 minutes before taking PrEP |
| | | 2. SMS reminder | A telephone call made to a mobile phone 30 minutes before taking PrEP |
| | | 3. None | No reminder |

SMS: Short Message Service

**Table 2. DCE choice sets with neither option.**

| Choice set | Alternative | Place of PrEP Delivery | PrEP Provider | Delivery Channel | Additional support |
|---|---|---|---|---|---|
| 1 | I | Hotspot | None | Peer | SMS reminder |
|  | II | Home | Peer | In-person | call reminder |
|  | III | Community | Peer | Family member | SMS reminder |
|  | IV | Neither | | | |
| 2 | I | Home | HW | Family member | call reminder |
|  | II | Community | Peer | Family member | SMS reminder |
|  | III | Health Facility | None | In-person | SMS reminder |
|  | IV | Neither | | | |
| 3 | I | Hotspot | None | Family member | call reminder |
|  | II | Health Facility | HW | Peer | SMS reminder |
|  | III | Community | Peer | Family member | SMS reminder |
|  | IV | Neither | | | |
| 4 | I | Home | HW | In-person | SMS reminder |
|  | II | Community | None | Peer | call reminder |
|  | III | Community | Peer | Family member | SMS reminder |
|  | IV | Neither | | | |
| 5 | I | Health Facility | HW | Peer | call reminder |
|  | II | Community | Peer | Family member | SMS reminder |
|  | III | Hotspot | Peer | In-person | SMS reminder |
|  | IV | Neither | | | |
| 6 | I | Health Facility | Peer | In-person | call reminder |
|  | II | Hotspot | HW | Family member | None |
|  | III | Community | Peer | Family member | SMS reminder |
|  | IV | Neither | | | |
| 7 | I | Community | Peer | Family member | SMS reminder |
|  | II | Home | Peer | Peer | None |
|  | III | Hotspot | HW | Family member | call reminder |
|  | IV | Neither | | | |
| 8 | I | Community | Peer | Family member | SMS reminder |
|  | II | Hotspot | HW | In-person | SMS reminder |
|  | III | Home | Peer | Peer | call reminder |
|  | IV | Neither | | | |

The D-efficiency of the design was 88.4%, which is above the 80% recommended score

We developed visual aids for each choice set to accommodate individuals with limited literacy skills (S1 File). Data on age, marital status, education level, duration on PrEP, current method of obtaining PrEP, and underlying comorbidities were collected by trained research assistants experienced in quantitative research supervised by the Principal Investigator. Data was entered into Excel 2019 and exported to STATA version 17.0 for analysis.

## DCE design

We employed a mixed methods design to sequentially determine an optimal PrEP delivery model. The first step involved analyzing previously collected qualitative data on barriers and facilitators faced by FSWs in Uganda when starting and adhering to PrEP [20]. We used an inductive analytic approach to identify preferred attributes and desirable qualities of

a PrEP delivery model based on input from current PrEP users. The second step involved a systematic review of PrEP uptake and retention among FSWs using various delivery approaches [19]. The DCE design considered all factors that could impact the decision-making process regarding the choice of the PrEP delivery model because failing to include crucial attributes in the study could introduce bias into the results [23]. This mixed-method approach informed the creation of a comprehensive list of potential attributes and attribute levels that could influence the optimal hypothetical PrEP delivery model [19,20].

## Generation of choice sets

We utilized a fractional factorial (D-efficient) design to generate choice sets that were optimally balanced within the given constraints. This method was chosen due to the large number of variable attributes, each with varying levels. Additionally, we did pilot testing to reduce the number of hypothetical delivery models presented to participants and avoid respondent fatigue. To ensure accuracy and effectiveness, the pilot testing of the initial DCE utilized a "think aloud" approach guided by established best practice guidelines [24]. This allowed participants to verbalize their thought processes while responding to the survey, thus identifying unclear or confusing questions and addressing other potential issues [18]. The pilot phase further evaluated the attribute's reliability among our target population. It permitted the assessment of participants' comprehension and interpretation of the tasks and questions, and estimated the completion time for the survey. Based on feedback from the pilot, adjustments were made to attribute design, question phrasing, and overall survey structure to ensure accurate testing. The profiles identified by the experimental design were then grouped into choice sets that were presented to the 203 FSWs in the form of a questionnaire with three main sections: (i) an introduction of the purpose of the DCE and how to respond correctly; (ii) questions about the participants' socio-demographic characteristics that were expected to influence their preference for a particular PrEP delivery model; (iii) the choice sets. The DCE survey questionnaire is attached as S1 File.

## Data collection

The data for this study were initially collected through a paper-based (questionnaire) survey, which was later transcribed into Excel for further analysis. FSW filled out the questionnaires manually, with the assistance of research assistants, and later, trained data entry personnel entered the responses into Excel. This approach was beneficial as FSW had limited ability to use digital tools and preferred to use paper-based formats. Several key checks were implemented throughout the data entry process to ensure data quality. First, independent personnel entered data twice to reduce the risk of human error, with discrepancies flagged and reviewed for consistency by the principal investigator. Second, consistency checks were performed for responses that followed logical patterns, such as numeric ranges or categorical answers, to identify any outliers or invalid entries. Additionally, a subset of questionnaires was selected for manual spot checks, comparing the paper responses to the digital entries to identify any discrepancies. Finally, after the initial data entry, a cleaning process was carried out, which included checking for missing values, duplicates, or out-of-range values that might indicate errors. By combining manual checks, double data entry, and cross-verification, we significantly minimized potential errors and ensured the integrity of the dataset.

## Statistical analysis

We analyzed the data using STATA 17.0 (Stata Corp, College Station, Texas, USA). We used descriptive statistics to summarize participants' socio-demographic characteristics. A panel-data mixed logit model was employed to assess FSW preferences and attribute trade-offs, accounting for correlated choice sets and case-specific covariates such as age, education level, current PrEP model, duration of PrEP, and use of non-PrEP drugs. The PrEP delivery models were constructed using a backward elimination approach, with the opt-out option (selecting none of the alternative models) set as the baseline alternative. Models were compared using the Akaike Information Criteria (AIC) and Bayesian Information Criteria (BIC). The model

with the lowest AIC and BIC values was determined to be superior. The finalized model, which consisted of age, education level, and current PrEP delivery model as case-specific covariates, was selected based on these criteria. The number of simulations was increased from 300 to 1,000 to ensure robustness before the model was finalized. The "margin" command was utilized to calculate expected probabilities for selecting alternative PrEP delivery models.

Additionally, we evaluated the impact of increasing age and categorical variable levels on the probabilities of choosing these alternative models through contrasts. Subsequently, marginal utilities for each alternative PrEP delivery model were calculated using linear predictions from the final model. Finally, based on the highest median utility score and interquartile range (IQR), we identified our preferred option among the alternative models for delivering PrEP. Utility scores are numerical values that reflect how much "value," "preference," or "satisfaction" a person assigns to a particular outcome or choice, i.e., the PrEP delivery model. Utility scores (usually range between **0 and 1**). The higher the utility score, the greater the preference.

### Validity and reliability of the experiment

To ensure that the DCE accurately measured what it intended to measure and that the results were reliable and applicable, we consulted with experts on the list of attributes drawn from our prior qualitative research [20]. This process ensured that the attributes and levels included in the DCE were relevant to the decision-making process and covered all important aspects of the choice context. We also conducted a pilot test with a small sample of 20 FSWs to identify any issues with the design, such as confusing questions or unrealistic choice scenarios. We used the feedback from the pilot test to refine the DCE. We used the likelihood ratio test, AIC, and BIC to check how well the choice model fit the data.

### Data quality control

We monitored the duration of participant responses to ensure that the choices were being made thoughtfully. Responses completed significantly faster than average (within 5 minutes) were flagged for further review, as they indicated that the participant was not properly considering the options. To assess whether participants were paying attention, we included a set of identical choice sets within the questionnaire. This method helped identify if FSW were selecting the same option repeatedly without regard to the alternatives. FSWs who consistently made the same choice in repeat sets were flagged for potential invalidity. For straight-line responses, where FSW selected the same answer across all choice sets, we used data analysis to identify patterns that indicate a lack of engagement or random selection. If FSW responses appeared to follow a straight line or exhibited no variability, we treated this as a potential sign of invalidity and excluded such responses from the analysis.

### Ethics approval

The study was approved by the Makerere University School of Medicine Research Ethics Committee (Mak-SOMREC-2022–299) and the Uganda National Council for Science and Technology (SS1223ES). Administrative clearance was obtained from Makerere University's Clinical Epidemiology Unit and Mulago National Referral Hospital Ethics Committee. All participants provided written informed consent before participating in the study. They received an IRB-approved reimbursement of 20,000 Uganda Shillings ($5.30) for their time, effort, and transportation costs.

## Results

### Participant characteristics

The median age of study participants was 24 years (IQR 20, 32), and 97.5% (198/203) were single and living in Kampala. Forty-one percent (84/203) of FSWs received PrEP from health facilities. Another 41% accessed PrEP through community delivery, while 37% (75/203) had been on PrEP for over one year. Nearly two-thirds of the participants reported taking other medications in addition to their PrEP pills (**Table 3**).

**Table 3. Characteristics of study participants.**

| Variable | Categories | Frequency (N=203) (%) |
|---|---|---|
| Age | Median (IQR) | 24 (20, 32) |
| Marital status | Married | 5 (2.5) |
| | Not married | 198 (97.5) |
| Residence | Outside Kampala | 5 (2.5) |
| | Within Kampala | 198 (97.5) |
| Education level | No education | 12 (5.9) |
| | Primary level | 95 (46.8) |
| | Secondary level | 86 (42.4) |
| | Post-secondary | 10 (4.9) |
| Duration on PrEP | Less than six months | 84 (41.4) |
| | >6months-1 year | 44 (21.7) |
| | >1 year | 75 (37.0) |
| Current PrEP model | Facility | 84 (41.4) |
| | Community | 84 (41.4) |
| | Both | 35 (17.2) |
| Comorbidity | Present | 128 (63.0) |
| | Absent | 75 (37.0) |

## PrEP delivery model preferences and attributes

Model 4 (health facility/HCW/in-person/SMS) had the highest utility score of being chosen (0.867), followed by utility scores for model 2 (home/peer/in-person/phone call) (0.749), and model 3 (Home/HCW/CHW/phone call) (0.727), (Table 4). Still, model 4 (health facility/healthcare worker/in-person/short message service) was the preferred model for delivering PrEP services across the ages of 15–19, 20–24, and ≥25 years, with median utility scores of 0.88, 0.87, and 0.85, respectively (Table 5).

**Table 4. Utility scores of choosing PrEP delivery.**

| # | Model | Median utility | IQR |
|---|---|---|---|
| 1 | **Health facility/HCW/in-person/SMS** | **0.867** | **0.823, 0.941** |
| 2 | Home/peer/in-person/phone call | 0.749 | 0.646, 0.787 |
| 3 | Home/HCW/CHW/phone call | 0.727 | 0.605, 0.800 |
| 4 | Hotspot/HCW/CHW/phone call | 0.678 | 0.490, 0.715 |
| 5 | Home/HCW/in-person/SMS | 0.674 | 0.617, 0.748 |
| 6 | Health facility/peer/in-person/phone call | 0.579 | 0.470, 0.661 |
| 7 | Home/peer/in-person/None | 0.568 | 0.311, 0.674 |
| 8 | Health facility/HCW/peer/SMS | 0.558 | 0.471, 0.608 |
| 9 | Community/pharmacist/peer/SMS | 0.547 | 0.320, 0.591 |
| 10 | Health facility/HCW/peer/phone call | 0.504 | 0.431, 0.538 |
| 11 | Opt-out (None of the models) | 0.500 | 0.500, 0.500 |
| 12 | Hotspot/peer/in-person/SMS | 0.478 | 0.246, 0.561 |
| 13 | Community/HCW/CHW/None | 0.470 | 0.280, 0.526 |
| 14 | Hotspot/HCW/in-person/phone call | 0.460 | 0.237, 0.564 |
| 15 | Community/peer/in-person/ phone call | 0.456 | 0.438, 0.610 |
| 16 | Hotspot/HCW/in-person/SMS | 0.448 | 0.383, 0.629 |

**Table 5. Utility scores of PrEP delivery models by age groups.**

| PrEP delivery model | 15–19 years | 20–24 years | ≥25 years |
|---|---|---|---|
| | Median utility (IQR) | Median utility (IQR) | Median utility (IQR) |
| Health facility/HCW/in-person/SMS | **0.88 (0.84, 0.95)** | **0.87 (0.85, 0.94)** | **0.85 (0.80, 0.93)** |
| Home/peer/in-person/phone call | 0.73 (0.61, 0.76) | 0.74 (0.63, 0.77) | 0.78 (0.68, 0.82) |
| Home/HCW/CHW/phone call | 0.61 (0.56, 0.75) | 0.66 (0.59, 0.78) | 0.76 (0.67, 0.84) |
| Hotspot/HCW/CHW/phone call | 0.67 (0.48, 0.69) | 0.68 (0.49, 0.70) | 0.69 (0.50, 0.73) |
| Home/HCW/in-person/SMS | 0.67 (0.62, 0.74) | 0.67 (0.62,0.74) | 0.68 (0.62, 0.76) |
| Health facility/peer/in-person/phone call | 0.61 (0.59, 0.72) | 0.58 (0.55, 0.67) | 0.51 (0.43, 0.63) |
| Home/peer/in-person/None | 0.53 (0.27,0.62) | 0.57 (0.25, 0.62) | 0.63 (0.39, 0.71) |
| Health facility/HCW/peer/SMS | 0.61 (0.54, 0.68) | 0.59 (0.52, 0.61) | 0.49 (0.43, 0.58) |
| Community/pharmacist/peer/SMS | 0.53 (0.29, 0.55) | 0.53 (0.30, 0.57) | 0.58 (0.35, 0.63) |
| Opt-out (None of the models) | 0.50 (0.50, 0.50) | 0.50 (0.50, 0.50) | 0.50 (0.50, 0.50) |
| Health facility/HCW/peer/phone call | 0.50 (0.41, 0.50) | 0.51 (0.42, 0.52) | 0.52 (0.46,0.56) |
| Hotspot/peer/in-person/SMS | 0.46 (0.27,0.50) | 0.43 (0.29, 0.52) | 0.54 (0.33, 0.62) |
| Community/HCW/CHW/None | 0.44 (0.25, 0.49) | 0.46 (0.26, 0.50) | 0.51 (0.31, 0.58) |
| Hotspot/HCW/in-person/phone call | 0.43 (0.20, 0.49) | 0.43 (0.22, 0.53) | 0.52 (0.28, 0.64) |
| Community/peer/in-person/ phone call | 0.56 (0.44, 0.61) | 0.53 (0.44, 0.61) | 0.46 (0.44, 0.60) |
| Hotspot/HCW/in-person/SMS | 0.35 (0.34, 0.37) | 0.40 (0.38, 0.42) | 0.55 (0.46, 0.67) |

The study participants who had previously experienced health facility and community-based PrEP delivery models generally preferred the health facility/healthcare worker/in-person/short message service model. However, those who had not utilized both health facility and community-based models simultaneously showed a greater preference for the hotspot/healthcare worker/community health worker/phone call model, although their perceived utility was similar to that of the health facility/healthcare worker/in-person/short message service model (median utility scores 0.94 and 0.81, respectively) (**Table 6**).

**Table 6. Utility scores by the current PrEP model being utilized by FSW.**

| PrEP delivery model | Health Facility model | Community model | Both health facility &community |
|---|---|---|---|
| | Median utility (IQR) | Median utility (IQR) | Median utility (IQR) |
| Health facility/HCW/in-person/SMS | **0.94 (0.94, 0.95)** | **0.81 (0.79, 0.85)** | 0.86 (0.84, 0.88) |
| Home/peer/in-person/phone call | 0.64 (0.61, 0.68) | 0.77 (0.75, 0.81) | 0.79 (0.76, 0.82) |
| Home/HCW/CHW/phone call | 0.66 (0.60,0.73) | 0.81 (0.78, 0.85) | 0.50 (0.43, 0.60) |
| Hotspot/HCW/CHW/phone call | 0.49 (0.47, 0.50) | 0.69 (0.68, 0.71) | **0.88 (0.87, 0.89)** |
| Home/HCW/in-person/SMS | 0.59 (0.55, 0.62) | 0.70 (0.67, 0.74) | 0.80 (0.76, 0.81) |
| Health facility/peer/in-person/phone call | 0.47 (0.43, 0.55) | 0.60 (0.55, 0.67) | 0.72 (0.66, 0.75) |
| Home/peer/in-person/None | 0.29 (0.25, 0.36) | 0.67 (0.61, 0.71) | 0.62 (0.56, 0.69) |
| Health facility/HCW/peer/SMS | 0.52 (047, 0.58) | 0.62 (0.58, 0.67) | 0.42 (0.38, 0.47) |
| Community/pharmacist/peer/SMS | 0.31 (0.29, 0.35) | 0.57 (0.55, 0.61) | 0.60 (0.57, 0.64) |
| Health facility/HCW/peer/phone call | 0.43 (0.42, 0.45) | 0.52 (0.51, 0.53) | 0.58 (0.57, 0.60) |
| Opt-out (None of the models) | 0.50 (0.50, 0.50) | 0.50 (0.50, 0.50) | 0.50 (0.50, 0.50) |
| Hotspot/peer/in-person/SMS | 0.30 (0.26, 0.35) | 0.56 (0.52, 0.62) | 0.51 (0.46, 0.57) |
| Community/HCW/CHW/None | 0.27 (0.25, 0.30) | 0.50 (0.48, 0.55) | 0.54 (0.52, 0.60) |
| Hotspot/HCW/in-person/phone call | 0.23 (0.19, 0.28) | 0.56 (0.50, 0.62) | 0.53 (0.47, 0.61) |
| Community/peer/in-person/ phone call | 0.46 (0.45, 0.61) | 0.44 (0.44, 0.60) | 0.68 (0.53, 0.68) |
| Hotspot/HCW/in-person/SMS | 0.42 (0.37, 0.53) | 0.42 (0.37, 0.51) | 0.78 (0.72, 0.84) |

**Table 7. Utility scores of PrEP delivery models by education level.**

| PrEP delivery model | No education | Primary education | Secondary education | Post-Secondary education |
|---|---|---|---|---|
| | Median utility (IQR) | Median utility (IQR) | Median utility (IQR) | Median utility (IQR) |
| Health facility/HCW/in-person/SMS | **0.85 (0.83, 0.91)** | **0.67 (0.55, 0.68)** | **0.86 (0.80, 0.94)** | 0.83 (0.78, 0.93) |
| Home/peer/in-person/phone call | 0.63 (0.57, 0.67) | 0.74 (0.65, 0.77) | 0.77 (0.65, 0.79) | 0.87 (0.79, 0.89) |
| Home/HCW/CHW/phone call | 0.62 (0.44, 0.73) | 0.52 (0.45, 0.59) | 0.68 (0.61, 0.81) | 0.75 (0.72, 0.84) |
| Hotspot/HCW/CHW/phone call | 0.42 (0.28, 0.58) | 0.67 (0.49, 0.69) | 0.70 (0.50, 0.75) | 0.68 (0.48, 070) |
| Home/HCW/in-person/SMS | 0.72 (0.63, 0.76) | 0.59 (0.51, 0.67) | 0.74 (0.62, 0.76) | **0.87 (0.79, 0.87)** |
| Health facility/peer/in-person/phone call | 0.58 (0.40, 0.69) | 0.59 (0.51, 0.67) | 0.50 (0.44, 0.60) | 0.69 (0.63, 0.76) |
| Home/peer/in-person/None | 0.76 (0.59, 0.81) | 0.63 (0.35, 0.69) | 0.57 (0.25, 0.62) | 0.43 (0.16, 0.46) |
| Health facility/HCW/peer/SMS | 0.47 (0.40, 0.61) | 0.44 (0.44, 0.53) | 0.59 (0.50, 0.64) | 0.60 (0.58,0.67) |
| Community/pharmacist/peer/SMS | 0.39 (0.27, 0.46) | 0.89 (0.85, 0.95) | 0.56 (0.32, 0.60) | 0.65 (0.40, 0.70) |
| Health facility/HCW/peer/phone call | 0.46 (0.44, 0.53) | 0.44 (0.44, 0.530) | 0.51 (0.43, 0.57) | 0.48 (0.41, 051) |
| Opt-out (None of the models) | 0.50 (0.50, 0.50) | 0.45 (0.23, 0.54) | 0.50 (0.50, 0.50) | 0.50 (0.50, 0.50) |
| Hotspot/peer/in-person/SMS | 0.22 (0.17, 0.25) | 0.48 (0.30, 0.54) | 0.51 (0.31, 0.58) | 0.62 (0.39, 0.63) |
| Community/HCW/CHW/None | 0.51 (0.40, 0.59 | 0.45 (0.27, 0.49) | 0.50 (0.28, 0.54) | 0.61 (0.39, 0.67) |
| Hotspot/HCW/in-person/phone call | 0.36 (0.26, 0.45) | 0.45 (0.44, 0.54) | 0.54 (0.25, 0.61) | $8.7 \times 10^{-8}$ ($2.3 \times 10^{-8}$, $1.0 \times 10^{-7}$) |
| Community/peer/in-person/ phone call | 0.43 (0.41, 0.47) | 0.45 (0.27, 0.49) | 0.61 (0.60, 0.62) | $7.7 \times 10^{-8}$ ($7.2 \times 10^{-8}$, $7.7 \times 10^{-8}$) |
| Hotspot/HCW/in-person/SMS | 0.18 (0.10, 0.34) | 0.44 (0.37, 0.57) | 0.45 (0.40, 0.68) | 0.71 (0.68, 0.76) |

The preferred PrEP delivery model among participants with no primary and secondary education levels was health facility/healthcare worker/in-person/short message service (utility score 0.86). However, post-secondary education participants preferred the health facility/healthcare worker/peer/short message service model (utility score 0.83). This choice closely aligned with participants' preferences in other education categories (**Table 7**).

## Discussion

This discrete choice experiment, conducted with cisgender FSWs in Kampala, Uganda, utilized a cross-sectional survey to determine the most end-user-centric PrEP delivery model. FSW preferred delivery through a health provider at the sex worker-friendly MARPI clinic, with additional support in the form of SMS reminders for enhancing PrEP adherence and engagement in care. These findings provide valuable insights into the preferences of FSWs for PrEP delivery, emphasizing the importance of provider involvement and tailored support in improving access and uptake within this key population. Findings from the qualitative phase revealed that healthcare providers at the clinic were perceived as friendly and non-judgmental, creating a comfortable environment for these women [20]. They were also seen as capable of maintaining confidentiality while delivering PrEP services. Participants reported feeling welcomed, included, and safe at the MARPI clinic, which encouraged them to continue utilizing its services [19,20].

The assumption of accurate and complete information is similar to findings of prior studies that FSWs appreciate PrEP introduction within familiar and trusted "friendly" clinics tailored for sex workers and value positive encouragement from clinic staff [25–27]. The plausibility of this assumption is that, whereas FSW might not have had accurate and complete information on all the PrEP delivery models, our DCE mitigated this issue by focusing on key attributes, hence allowing FSW to make choices based on the most relevant factors. In our study, healthcare workers were perceived as knowledgeable individuals with specialized knowledge about PrEP. This preference for healthcare workers as PrEP providers is consistent with previous research conducted in Uganda, which found that PrEP delivery based at health facilities required healthcare providers to have sufficient knowledge and confidence in discussing antiretroviral medications for HIV prevention with clients [28]. Similar results have been demonstrated in family health programs in Brazil, Bangladesh, and Nepal,

in which health workers positively influenced health by serving as entry points, bridges, and connectors to healthcare services, systems, and resources [29–31]. Our results emphasize the importance of involving healthcare workers in PrEP delivery for FSWs [20].

Whereas most FSWs preferred healthy facility-based PrEP delivery, some preferred community-based healthcare services that involve community providers and peers rather than solely facility-based options, as this can help overcome stigma and discrimination barriers [32]. This divergence in preference contradicts the DCE homogenous assumption across the sample that preferences among individuals can vary greatly by previous experience. The plausibility of this assumption is limited to diverse populations. The World Health Organization recommends differentiated approaches for delivering PrEP services, prioritizing the individual and community [33]. These approaches are adapted to the specific needs and preferences of individuals who may benefit from PrEP. Implementing these differentiated services can improve PrEP acceptability and accessibility and support its ongoing use and effectiveness [34]. Community-based delivery options such as pharmacies, community organizations, drop-in centers, and mobile clinics complement facility-based care by providing strong linkages and referral pathways for those seeking treatment [35]. Our findings underscore the importance of customizing differentiated service delivery models for FSWs.

Research has demonstrated the effectiveness of SMS interventions in promoting medication adherence. Clients who received SMS reminders reported higher treatment adherence than those who did not [36,37]. This may be attributable to the non-intrusive nature of text message reminders compared to other adherence strategies [38]. Furthermore, the simplicity and user satisfaction associated with such reminders make them a valuable tool in healthcare services. A study on improving medication adherence among type 2 diabetes patients through SMS reminders showed that this method was relatively simple and had minimal impact on daily routines [39]. This review concluded that text messages increased adherence and improved health outcomes [37,40], although there were limitations for those without access to a phone or reliable electricity.

### Strengths and limitations

Our study has several strengths. We conducted a DCE in a busy health facility that provides comprehensive HIV care and prevention services to more than 80% of FSWs in Kampala. This approach allowed for a significant representation of FSWs' preferences regarding PrEP delivery. Additionally, our results are based on a robust sample size of >200 FSWs, surpassing the recommended number of 150 for DCEs that has sufficient statistical power to detect significant effects [23]. We deliberately selected FSWs taking PrEP for at least two months, ensuring their preferences were grounded in their firsthand experiences with current delivery methods. Additionally, we used pictograms to aid participants' understanding of choice sets and minimize strategic biases that may have skewed preferences. However, our study had limitations. Notably, our sample did not include pregnant FSWs despite being at twice the risk for HIV acquisition during pregnancy and postpartum compared to non-pregnant periods. The other limitation is that DCEs are theoretical in offering choices. Therefore, randomized controlled trials should be done as part of person-centered care to assess the effectiveness of various model preferences in prevention coverage. Future studies should address this gap and include pregnant FSWs and other vulnerable populations to gain a more comprehensive understanding of their perspectives.

### Conclusions and recommendations

FSWs showed a clear preference for receiving PrEP directly from a health worker within a friendly and supportive healthcare setting. To enhance adherence and mitigate the risk of adverse health outcomes like HIV acquisition due to non-adherence, there is a significant need for supplementary support mechanisms. Implementing SMS reminders could improve adherence rates and ensure better health outcomes for FSWs using PrEP. These preferences should be considered when designing future approaches for delivering PrEP through the National PrEP Program. Optimization of health worker training and retention in key population programming should be prioritized as a key component of health system

improvements. This should include advocating for hiring health workers to effectively implement and expand programs nationwide, ensuring broader access to services for FSWs and enhancing the overall impact of such initiatives. As mentioned above, further evaluation of the feasibility and effectiveness of this PrEP delivery model is necessary, including in vulnerable populations, such as pregnant FSWs. A study is underway to determine the feasibility and acceptability of this preferred PrEP delivery model for FSWs. A study is underway to determine the feasibility and acceptability of our preferred PrEP delivery model for FSWs.

## Supporting information

**S1 File. DCE survey questionnaire.**
(PDF)

**S1 Data. DCE data set.**
(XLSX)

## Author contributions

**Conceptualization:** Ruth Mpirirwe, Mayanja M. Kajumba, Joan Kalyango, Charles Karamagi, Agnes Kiragga, Mari Armstrong-Hough, Anne R. Katahoire, Moses R Kamya, Andrew Mujugira.

**Data curation:** Ruth Mpirirwe, Andrew Mujugira.

**Formal analysis:** Ruth Mpirirwe, Laban Muteebwa, Onesmus Kamacooko, Felix Wamono, Mayanja M. Kajumba, Katumba James Davis, Andrew Mujugira.

**Funding acquisition:** Ruth Mpirirwe, Moses R Kamya.

**Investigation:** Ruth Mpirirwe.

**Methodology:** Ruth Mpirirwe, Rita Makabayi-Mugabe, Laban Muteebwa, Felix Wamono, Andrew Mujugira.

**Project administration:** Ruth Mpirirwe, Joan Nangendo, Fred C. Semitala, Peter Kyambadde.

**Resources:** Ruth Mpirirwe, Anne R. Katahoire, Moses R Kamya.

**Software:** Ruth Mpirirwe, Onesmus Kamacooko.

**Supervision:** Ruth Mpirirwe, Joan Kalyango, Charles Karamagi, Anne R. Katahoire, Moses R Kamya, Andrew Mujugira.

**Validation:** Ruth Mpirirwe, Moses R Kamya, Andrew Mujugira.

**Visualization:** Ruth Mpirirwe, Andrew Mujugira.

**Writing – original draft:** Ruth Mpirirwe, Andrew Mujugira.

**Writing – review & editing:** Ruth Mpirirwe, Sarah EG Moor, Anne R. Katahoire, Moses R Kamya, Andrew Mujugira.

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
