## [Decision Letter · Decision Letter 0]

PGPH-D-24-02264

Female Sex Worker Preferences for HIV Pre-Exposure Prophylaxis Delivery in Uganda: A Discrete Choice Experiment

Dear Dr. Mpirirwe,

Thank you for submitting your manuscript to PLOS Global Public Health. After careful consideration, we feel that it has merit but does not fully meet PLOS Global Public Health’s publication criteria as it currently stands. Therefore, we invite you to submit a revised version of the manuscript that addresses the points raised during the review process.

We look forward to receiving your revised manuscript.

Kind regards,

Nazmul Alam, MPH, DrPH

Academic Editor

Journal Requirements:

Reviewers' comments:

**Comments to the Author**

1. Does this manuscript meet PLOS Global Public Health’s publication criteria?

Reviewer #1: Yes

Reviewer #2: No

2. Has the statistical analysis been performed appropriately and rigorously?

Reviewer #1: Yes

Reviewer #2: No

3. Have the authors made all data underlying the findings in their manuscript fully available (please refer to the Data Availability Statement at the start of the manuscript PDF file)?

Reviewer #1: Yes

Reviewer #2: No

4. Is the manuscript presented in an intelligible fashion and written in standard English?

Reviewer #1: Yes

Reviewer #2: Yes

Reviewer #1: Well written and informative article. I have learned a lot from it.

Minor revision: One of the two major findings is the clear preference of FSWs for receiving PrEP from competent and supportive health workers. I was unable to get a sense from the article whether there are workforce challenges such as shortages or maldistribution in Uganda in particular the workforce that provides such care for FSWs. I agree with the authors emphasizing the need for supplementary support mechanisms such as implementing SMS reminders. However, optimization of the workforce in terms of training and retention is not discussed or put forth as a recommendation. From a health system perspective, this is an excellent opportunity to advocate for upscaling the workforce in order to implement more such programs throughout the country to serve more FSWs. I suggest adding this to the discussion and recommendations.

Reviewer #2: Thank you for the opportunity to review this manuscript; it is an exciting exploration of client-centered prevention for a highly exposed population. Beyond the high importance and innovation of this work, a key strength is the substantial preliminary research around attributes and levels to inform the DCE. I raise below points of confusion or weakness for further consideration.

1) Target population and validity of findings. There is a discordance between the stated aims / interpretation of the findings and the sample collected. The sample collected includes PrEP-experienced FSW provided care at a specialty clinic, but the paper as framed as applicable to FSW in Uganda more broadly. The difference is highly relevant in considering people's preferences for health services - the actual sample is highly selected based on these preferences, and the experience at the clinic likely in turn informs future preferences. While the introduction does not provide specific numbers for proportion of FSW in Uganda trying PrEP and staying on PrEP, it does state that PrEP use has been fairly low and persistence quite low. This suggests that the sample analyzed, which includes many long-term PrEP users, is likely to differ from the overall population of FSW in key ways related to health service use. The reliance on MARPI similarly will distinguish the sample from a broader target population in that experiences there are quite positive (per the Discussion section) in ways that 'preference for a clinic and HCW delivery' could obfuscate - preferences for clinics and HCW could be quite different for those picturing a typical public clinic. I would strongly suggest reframing the study objective and the interpretation of the finding to reflect a primary interest in supporting PrEP persistence, which fits with the choice of limiting the sample to PrEP-experienced FSW. The potential generalizability of these findings to FSW who have not already overcome barriers to PrEP and/or who don’t have access to MARPI should be a major focus of the discussion section.

2) DCE design: the selected attributes and levels may not directly violate the assumption of independence, but they certainly give rise to some convoluted options, like receiving PrEP from a peer at a health facility, or a health care worker giving PrEP to a peer to provider to the respondent. The presentation in the attached survey file seems to exacerbate this confusion, labeling the choices based on one attribute and leaving issues like how the PrEP picked up by a peer from a pharmacist actually gets to the respondent unanswered. (Other issues like referring to pharmacist in some settings and HCW in others, while these are the same level in the underlying design, are also concerning.) The ultimate design and certainly the analytic approach seem to move this experiment away from a traditional DCE, where each attribute is independent and assessed as its own contributor towards preference, and towards a conjoint analysis of overall models rather than separate attributes. Please provide further description and justification of this approach, including evidence from the preliminary assessments that respondents were able to assess each of the 4 attribute-levels within the choice sets as presented. Alternatively, please provide the theoretical basis or other reference material for this as a distinct type of preference experiment, with choice task presentation and analytic methods defined accordingly.

3) Analysis: as mentioned in the prior point, the analysis presented diverges from traditional DCE analysis, which estimates utilities for each attribute (for the full sample and/or classes therein) in order to quantify the relative importance of the attributes and the relative preference for each level. It is then possible to estimate uptake of specific models combining attribute levels. The current approach jumps directly to assessment of pre-defined combination of attribute levels in a way that is not fully described, such that I am unable to assess its rigor and interpretability. What was the full set of models tested - every permutation of the 4 attributes / 13 levels? And then a subset of them was selected to be shown here based on some sort of ‘backwards selection’? How are readers to interpret these utility values? There is not enough detail to understand what was down and how to interpret it. It is striking that the authors mention in the limitations section that the opt out was the single most selected option, but these results are not presented in the Results section itself and would be impossible to read from the Tables shown, where the opt out option is fixed at 0.50 as the reference value. The analysis I would expect to see would describe the counts of respondents (including quality checks, see below) and then provide an overall or by-class utility model to provide the relative utility of each level and the heterogeneity of these preferences. The importance of each attribute would be quantified so that we can better understand even in context of these overall models weather location vs. reminder vs. person is primarily driving choices. Then specific models like those used here could be tested to estimate relative preference or expected uptake. It’s possible the process presented here did address some of these steps, but it’s not possible for me to really understand it at present, and overall it does not seem to match the design process of creating a DCE with distinct attributes and levels.

Moderate:

The order of the methods section is a little jumbled - the information on the selection of attributes and levels is presented twice, as is the generation of choice sets, with information on data collection sandwiched between, and with more details on second presentation. I recommend streamlining for greater clarity.

The imbalance in the design is a little surprising - for reminder option, SMS appears 14 times, phone call 7, and no reminder appears only twice. Was this programmed in based on assumptions about preference or purely random? Did the pilot feedback include estimated preference levels that were used to re create the design matrix? What software was used for the design generation?

What was the mode of data collection, paper prior to the entry into Excel? Or collected on tablet / laptop directly into Excel? Please specify this and note any data quality checks, given that these are not standard survey collection tools with built in quality checks

What DCE quality checks were built in, regarding timing, repeat choice, internal validity? What tests were performed for straight-line responses or other signs of invalid responses? If none, please note as a limitation.

Please consider the DCE assumptions in the discussion section and assess the evidence and plausibility of these assumptions.

The first paragraph of the discussion refers more to the previous studies than this study. Please provide a succinct summary of the major insights of the current study before placing them in perspective later in the discussion section.

The Discussion section mentions 150 as a standard DCE size without citation; DCE sample size is fundamentally dependent on the design, so reference to a standard sample size is not appropriate.

Minor

Note that the rows for Table 1. 4. Additional support are flipped (phone call vs. text)

It’s difficult to interpret ‘none’ within the attributes - it’s defined as no preference between them, but on its face would be read as a preference against both, so ‘no provider’, e.g. self service, and no reminder. I would revise Table 1 to indicate that it means (at least based on the attached tool) ‘no provider’ ‘no reminder’ rather than no preference.

**Do you want your identity to be public for this peer review?** For information about this choice, including consent withdrawal, please see our Privacy Policy

Reviewer #1: **Yes: ** Professor Nazik Hammad

Reviewer #2: No

---

## [Editor Report · Decision Letter 1]

Female Sex Worker Preferences for HIV Pre-Exposure Prophylaxis Delivery in Uganda: A Discrete Choice Experiment

PGPH-D-24-02264R1

Dear Dr Mpirirwe,

We are pleased to inform you that your manuscript 'Female Sex Worker Preferences for HIV Pre-Exposure Prophylaxis Delivery in Uganda: A Discrete Choice Experiment' has been provisionally accepted for publication in PLOS Global Public Health.

Best regards,

Nazmul Alam, MPH, DrPH

Academic Editor